# A Prospective Study on Obstructive Sleep Apnea, Clinical Profile and Polysomnographic Variables

**DOI:** 10.3390/jpm13060919

**Published:** 2023-05-30

**Authors:** Geetha Kandasamy, Tahani Almeleebia

**Affiliations:** Department of Clinical Pharmacy, College of Pharmacy, King Khalid University (KKU), Abha 61421, Saudi Arabia

**Keywords:** sleep apnea, polysomnography, apnea hypopnea index, excessive daytime sleepiness

## Abstract

**Background:** Obstructive sleep apnea (OSA) is characterized by recurring episodes of pharyngeal collapse, which can partially or completely block airflow during sleep and cause cardiorespiratory and neurological imbalances. Therefore, the purpose of this study was to assess OSA and the relationship between AHI and polysomnographic characteristics in OSA patients. **Methodology:** A prospective study was conducted at the Department of Pulmonology and Sleep Medicine for two years. All 216 participants underwent polysomnography, and 175 of them were reported to have OSA (AHI ≥ 5), while 41 of them did not (AHI < 5). ANOVA and Pearson’s correlation coefficient test were performed. **Results:** In terms of the study population’s average AHI, Group 1 had 1.69 ± 1.34, mild OSA had 11.79 ± 3.55, moderate OSA had 22.12 ± 4.34, and severe OSA was found to have 59.16 ± 22.15 events/hour. The study group’s average age was 53.77 ± 7.19 out of 175 OSA patients. According to AHI, the BMI for mild OSA was 31.66 ± 8.32 kg/m^2^, for moderate OSA, it was 30.52 ± 3.99 kg/m^2^, and for severe OSA, it was 34.35 ± 8.22 kg/m^2^. The average number of oxygen desaturation events and snoring duration were 25.20 ± 18.63 and 24.61 ± 28.53 min, respectively. BMI (*r* = 0.249, *p* < 0.001), average oxygen saturation (*r* = −0.387, *p* < 0.000), oxygen desaturation (*r* = 0.661, *p* < 0.000), snoring time (*r* = 0.231, *p* < 0.002), and the number of snores (*r* = 0.383, *p* < 0.001) were the polysomnographic variables that showed significant correlations with AHI in the study group. **Conclusions:** In this study, a substantial prevalence of obesity and a high OSA frequency were found in men. Our research showed that individuals with obstructive sleep apnea experience nocturnal desaturations. Polysomnography is the primary test for early detection of this treatable condition.

## 1. Introduction

Obstructive sleep apnea (OSA) is characterized by recurring episodes of pharyngeal collapse, which can partially or completely block airflow during sleep and cause cardiorespiratory and neurological imbalances. Disruptive snoring, repeated episodes of total or partial pharyngeal obstruction, and excessive daytime sleepiness (EDS) are all linked to nocturnal hypoxemia. Although it can affect women and children, OSA is more frequently found in elderly males. According to studies, 10 to 15 percent of females and 15 to 30 percent of males in North America have OSA. According to reports, 5 percent of females and 15% of males are affected [1]. Between the ages of 30 and 69, 936 million individuals globally have mild to severe OSA, and 425 million people have moderate to severe OSA according to estimates based on 5 or more incidents each hour. The incidence of OSA differs by race as well. Independent of body weight, OSA is more common in African Americans under the age of 35 than in White Americans in the same age group. Despite lower rates of obesity, OSA prevalence in Asia is identical to that in the US [2].

Following menopause, the incidence increases to the point where postmenopausal people’s rates are the same [3]. It also causes behavioral abnormalities and affects neurocognition during the day [4]. Sudden variations in lung pressures and volumes result in respiratory acidosis, oxygen desaturation, sympathetic nervous system activation, and an imbalance in the phases of sleep [5]. A decrease in energy and general mental and physical health can come from OSA, which causes episodes of apneas and hypopneas to break the sleep stages. Combining these risk factors makes patients more likely to experience morbidity and mortality [6].

Obstructive sleep apnea (OSA) is characterized by at least five instances of interrupted breathing per hour of sleep [7]. It is estimated that more than 85% of people with clinically significant OSA have never received a diagnosis despite the fact that OSA is a rather prevalent medical illness [8]. This is assumed to reflect the fact that many patients with symptoms of OSA are not conscious of their severe snoring and nocturnal arousals. The main characteristics of OSA are indicators of sleep disturbance, including snoring and restlessness, interruptions of normal breathing patterns while sleeping, and daytime symptoms, such as exhaustion or difficulty concentrating, that are caused by nighttime sleep disturbances [9].

Sleep disruption is correlated with various daytime symptoms, such as daytime sleepiness, fatigue, and poor concentration. Furthermore, sufferers of OSA face higher risks of obesity and diabetes, as well as serious complications, such as cardiovascular and cerebrovascular events [10,11]. A recent analysis estimated that 936 million individuals of both genders aged 30–69 years were found to have obstructive sleep apnea worldwide [12].

Certain physical characteristics that may contribute to OSA include obesity, thickened lateral pharyngeal walls, nasal congestion, enlarged uvula, facial malformations, micrognathia, macroglossia, and tonsillar hypertrophy. Obesity contributes to airway narrowing through fatty infiltration of the tongue, soft palate, or other areas surrounding the airway [7]. The results of repeated brief oxygen saturation cycles followed by quick reoxygenation have also been studied. Hypoxemia impairs the production of nitrous oxide, a powerful vasodilator, directly affecting the vascular beds. Additionally, periods of hypoxia trigger the activation of numerous inflammatory cells that are potentially harmful to endothelial cells and predispose individuals to the formation of atherosclerotic plaques [13].

The nasopharynx’s muscles start to relax as the patient drifts off to sleep, and the surrounding tissue compresses, compromising the airway. The patient is awakened from sleep as carbon dioxide levels in the body begin to rise and oxygen levels decline. This results in an increase in sympathetic tone and a contraction of the nasopharyngeal tissue, which relieves the obstruction. However, once the patient falls asleep, the airway narrows once more until they are once again roused [14]. Polysomnography is used to confirm the diagnosis of OSA with an apnea hypopnea index (AHI) >5 per hour considered diagnostic. It is necessary to perform overnight supervised laboratory polysomnography (PSG) to confirm the diagnosis and better prioritize treatment options [15]. The greatest risk factor for OSA is obesity, and other characteristics, including an altered body mass index (BMI), a larger neck or waist, or a higher waist-to-hip ratio, are all thought to be risk factors. However, it seems that this link varies depending on social, environmental, and racial factors. Asians are more susceptible to obesity than Caucasians [16]. It is also becoming more widely recognized that craniofacial morphology plays a key role in the pathophysiology of OSA [17]. Therefore, the purpose of this study was to assess OSA and the relationship between AHI and polysomnographic characteristics in OSA patients.

## 2. Methodology

### 2.1. Study Design and Duration

The Department of Pulmonology and Sleep Medicine at the Multispeciality Hospital in India conducted a prospective study. The typical study period lasts two years.

### 2.2. Inclusion Criteria

Individuals who met the criteria for obstructive sleep apnea (OSA) and had not previously received therapy were registered. These criteria included excessive daytime sleepiness, upper airway blockage, loud snoring, shallow breathing, and breathing trouble. All patients had polysomnography studies performed.

### 2.3. Exclusion Criteria

This study excluded patients who could not afford to have the polysomnography done. Individuals with co-morbidities, terminal illnesses, central sleep apneas, and pediatrics were excluded.

### 2.4. Study Population

Based on the inclusion/exclusion criteria, we planned to recruit patients over the research period. During the study period, 1272 patients underwent a sleep study. Among 1272, 216 patients met our inclusion/exclusion criteria. Out of 216 patients, 175 were obstructive sleep apnea patients (OSA), and 41 were Non-OSA.

### 2.5. Patients and Predictive Risk Factors

All 216 participants underwent polysomnography, and 175 of them were reported to have OSA (AHI ≥ 5), while 41 of them did not (AHI < 5). Non-OSA patients with AHI < 5 and compliance were categorized into Group 1 (*n* = 41), while the remaining 175 subjects with AHI ≥ 5 were categorized into Group 2 (*n* = 175). The 175 patients in Group 2 were then divided into 3 subgroups based on AHI (events/hour).

Through the patients’ case reports and interactions, information about the patient’s demographics, family history of sleep disorders, cause for hospital admission, sleep history, snoring behaviors, and social habits were gathered. Weight in kg/height in meters squared was used to compute BMI. The Institutional Review Board of Multispeciality Hospital India gave its approval for this study’s methodology.

### 2.6. Polysomnography (PSG) Sleep Study

Using a recording device, common electrodes, and sensors, overnight polysomnography was carried out. Electroencephalograms (EEG), electro-oculograms (EOG), electrocardiograms (ECG), electromyograms (EMG) of the chin and legs, tracheal breath sounds, abdominal and thoracic wall movements, transcutaneous oxygen saturation, and body position were among the many variables tracked. Each individual underwent PSG for a minimum of six hours. The computer’s sleep records were rated for apneas, hypopneas, and sleep phases. Oronasal airflow ceasing for more than 10 s was considered to be an apnea. When there was no airflow but there were breathing efforts, the condition was called obstructive apnea. Hypopnea was defined as a decrease in respiratory airflow during a previous period of normal breathing for more than 10 s followed by a drop of at least 4% in oxygen saturation.

### 2.7. Statistical Analysis 

The results of the current investigation were interpreted using the following statistical methods. Data were examined using SPSS version 12.0. Findings are presented using descriptive statistics and percentages. ANOVA was performed to determine whether there were any statistically significant differences between the groups based on demographic and polysomnographic data. The Pearson’s correlation coefficient test was performed to examine the relationship between clinical and polysomnographic variables and the apnea hypopnea index.

## 3. Results

Throughout a two-year period, this prospective investigation was carried out. This study included 216 study participants who visited the Multispeciality Hospital’s Pulmonology Department and completed the polysomnography test. Values for each patient’s apnea hypopnea index (AHI) were collected from a polysomnography investigation. Study participants who had an apnea hypopnea index of less than five were classified as the Non-OSA Group 1, while those with an AHI of five or more were classified as OSA patients.

From this study, 175 patients with AHI values over 5 were classified as OSA Group 2, while 41 subjects with AHI values under 5 were classified as Non-OSA Group 1. Comparisons were made between the baseline values of Group 1 and Group 2 participants.

This study group’s average age was 53.77 ± 7.19. Males had an average age of 55.41 ± 11.24, while females had a median age of 52.1 ± 13.12 years. According to demographic information, there were 175 patients, 143 of whom were male (81.7%) and 32 of whom were female (18.3%). The majority (*n* = 25) of the 32 female patients were postmenopausal. The average age of the study population was 52.11 years for women and 55.41 years for men (Table 1). Group 2 (*n* = 175) was once more classified by the apnea hypopnea index as having mild OSA, moderate OSA, and severe OSA. Among the study population, 17.1% (*n* = 30) were classified as mild (AHI between 5 and 14.9), 34.3% (*n* = 60) as moderate (AHI between 15 and 29.9), and 48.6% (*n* = 85) as severe (AHI more than or equal to 30) (Table 2). In terms of the study population’s average AHI, Group 1 had 1.69 ± 1.34, mild OSA had 11.79 ± 3.55, moderate OSA had 22.12 ± 4.34, and severe OSA was found to have 59.16 ± 22.15 events/hour (Table 3). A total of 20% of the patients in the current study were smokers with 2.3% in the mild group, 8% in the intermediate group, and 9.7% in the severe group. Alcohol history revealed 31% of the population to be alcoholics, of whom 4% had mild OSA, 8.6% had moderate OSA, and 18.9% had severe OSA (Table 2). OSA is frequently characterized by snoring, breathing issues, and daytime sleepiness. Each patient’s symptoms were assessed, and 80% (*n* = 140) of them reported loud snoring, 56% (*n* = 98) had breathing issues, and 64% (*n* = 112) reported feeling sleepy during the day. Among patients with obstructive sleep apnea, 25.71 percent of the population had a family history of snoring (Table 4).

Obese people were found to have OSA more frequently. The average BMI for group 2 (OSA) was 32.17 ± 6.84 kg/m^2^ compared to 26.12 ± 2.72 kg/m^2^ for Group 1 (non-OSA). According to AHI, the BMI for mild OSA was 31.66 ±8.32 kg/m^2^, for moderate OSA, it was 30.52 ± 3.99 kg/m^2^, and for severe OSA, it was 34.35 ± 8.22 kg/m^2^ (Table 4). A total of 10.9% of OSA patients had an ideal BMI compared to the 29.7% who were overweight and 59.4% who were obese (Table 2).

The BMIs of Groups 2 and 1 differ significantly (*p* < 0.05) from those of Group 1. The study population’s groups with moderate and severe OSA experienced significant changes in BMI (*p* < 0.004). AHI and BMI were shown to be significantly correlated (*r* = 0.249, *p* < 0.05). (Table 5).

### Polysomnography Sleep Study

Eight hours were spent studying the patients’ sleep patterns using polysomnography. The study group’s mean oxygen saturation levels were 94.73 ± 2.91%, while Group 1′s values were 95.92 ± 4.15%, mild OSA was 95.66 ± 1.98%, moderate OSA was 95.22 ± 2.10%, and severe OSA was 92.12 ± 3.44%. The average number of oxygen desaturation events per hour for the study population was 25.20 ± 18.63, and the average number of events per hour for Group 1, mild, moderate, and severe groups were 6.19 ± 10.98, 21.22 ± 20.66, 23.38 ± 16.23, and 50.04 ± 26.67, respectively. The study group’s average snoring duration was 24.61 ± 28.53 min, while values for Group 1, the mild, moderate, and severe groups were 22.22 ± 39.61 min, 21.99 ± 25.51 min, 22.86 ± 16.49 min, and 31.39 ± 32.51 min, respectively. The average number of snoring episodes was 52.48 ± 61.34; the values for Group 1, mild, moderate, and severe were 35.68 ± 59.78, 41.49 ± 43.12, 46.33 ± 39.58, and 86.44 ± 102.88, respectively (Table 4).

BMI (*r* = 0.249, *p* < 0.001), average oxygen saturation% (*r* = −0.387, *p* < 0.000), oxygen desaturation events/hour (*r* = 0.661, *p* < 0.000), snoring time (*r* = 0.231, *p* < 0.002), and number of snores (*r* = 0.383, *p* < 0.001) were the polysomnographic variables that showed significant correlations with AHI in the study group. Table 5. Figure 1, Figure 2, Figure 3, Figure 4 and Figure 5.

## 4. Discussion

### 4.1. Prevalence and Age

The prevalence of OSA seems to gradually rise with middle age. This study’s participants were 53.76 ± 7.18 years old on average. Males were 55.41 ± 11.24 years old on average, and females were 52.11 ± 3.12 years old (Table 1). An earlier study found that men and people between the ages of 30–64 are more likely than women to suffer from OSA [18]. Another study found that people between the ages of 30 and 60 are more likely to experience respiratory problems when they sleep [1]. Our findings are also corroborated by Bixler et al. who discovered that the middle-aged population (ages 45 to 64) has the highest prevalence of OSA [19]. According to research by Ernst G. et al., OSA is very common in adults over 65 [20]. Moreover, according to Deng et al.’s research, age and obesity are significant risk factors for the severity of OSA [21].

### 4.2. Gender

The progression of OSA appears to be closely correlated with the male upper-body pattern of obesity. The male predominance of OSA may be influenced by gender differences in fat buildup, notably in the neck [22]. Our study’s findings indicate that 81.7% of participants were men, and 18.3% were women (Table 1). The current study definitely shows a significant prevalence of OSA in males. Our investigation revealed that, in line with the majority of other studies, men have more severe OSA than women [23]. It is unclear how the pathophysiologic mechanisms work. Variations in the upper airway’s dynamic characteristics could be extremely important. Elisa Perger et al. [24]. proposed that females’ upper airways were stiffer than men’, making females less prone to collapse and males more prone to pharyngeal collapse. The muscles that open the airways may also become more toned as a result of female hormones, reducing airway collapse. Female hormone levels rise to place a protective responsibility on the muscles that dilate the upper airways. According to reports, postmenopausal women are more likely than premenopausal women to suffer from sleep apnea [25]. Another study showed that postmenopausal women have a high prevalence of sleep apnea [26]. A total of 25 of the 32 female patients were postmenopausal, which confirms previous reports and shows a higher incidence.

### 4.3. Obesity

Obesity is a serious risk factor for the development of sleep apnea. This study makes it quite evident that the majority of OSA patients were morbidly obese. In this study, obesity was highly prevalent. Body mass index (BMI) ranged from 10.9% (*n* = 19) to 29.7% (*n* = 52) for optimum BMI, to 59.4% (*n* = 104) for overweight and obese.

OSA and obesity are closely related [27]. Obesity-related sleep apnea is caused by upper respiratory tract fat deposits in obese individuals [28]. The majority of the patients included in another study were overweight, confirming once more the substantial correlation between obesity and OSA. Moreover, at the time of the follow-up, we noticed a considerable decrease in BMI in the CPAP group [29]. These findings concur with those of Harsch et al. [30], suggesting that OSA patients who successfully treat their sleep apnea may also find it simpler to lose weight. With rising obesity, fat may begin to accumulate on the tongue and the soft tissue surrounding the pharynx. As a result, the pharyngeal airway appears to be shifting its obstruction to a lower level and becoming more collapsible while we sleep [31]. Moreover, our findings are consistent with earlier research that revealed obese OSA patients to have more severe cases of the condition [32]. The two main factors that predict sleep apnea are obesity and daytime sleepiness. Obesity is expected to affect 60 to 90 percent of people with sleep apnea [33]. Although it is one of the main risk factors, obesity has been shown to be the primary indicator of metabolic syndrome and a clinically significant cause of the severity of OSA [34]. Ciavarella D et al. concluded in their study that BMI has a significant effect on OSA severity [35].

In the current study, 20% of the patients smoked, and 31% of the participants had a history of alcohol abuse. According to this study, male obesity may play a role. Significant risk factors for OSA-related snoring include male gender, middle age, smoking, alcoholism, and physical infirmity [36]. Based on previous studies, individuals with more severe OSA and more common CVD comorbidities smoked more cigarettes per day, more packs per year, and consumed more Fagerstrom test for nicotine dependence [37]. In this investigation, we discovered that alcohol use was an independent risk factor for OSA and OSA with hypoxia and that after adjustment, alcohol use was strongly associated with AHI, particularly in females. It is recommended that people steer clear of alcohol and that drinkers abstain from drinking in order to lessen the risk and severity of OSA [38].

A total of 80% of the patients in the current study reported loud snoring, 56% had breathing problems, and 64% said they felt sleepy during the day when their symptoms were assessed. A family history of snoring was seen in 25.7 percent of patients with obstructive sleep apnea. According to Sowho M, the frequency and severity of snoring are related to the presence of OSA, indicating that higher levels of snoring increase the probability of developing OSA. Snoring severity and frequency are linked to OSA severity [39]. The most typical OSA symptoms according to Spicuzza L et al. include snoring and excessive daytime sleepiness. One of the primary symptoms of OSA, which is present in more than 80% of patients, is daytime sleepiness driven by nocturnal sleep fragmentation [40]. Even though people with FH of OSA are considerably younger, there is a higher chance of obesity, particularly central obesity [41]. A substantial increase in the likelihood of prevalent hypertension was found to be connected with OSA, defined as apneas and hypopneas marked by 3% desaturation episodes or arousals [42]. An earlier study revealed that middle-aged males have a significant rate of nocturnal desaturation events similar to our study [43]. It was discovered that individuals with OSA experienced severe daytime sleepiness due to poor nighttime oxygenation. This provided evidence that EDS and oxygen desaturation are closely related. Hypercapnia and hypoxia are also addressed by oxygen desaturation. The main cause of cardiovascular diseases was hypoxia followed by reoxygenation, which occurs repeatedly during the night and causes changes in perfusion, the production of free radicals, and oxygen desaturation in the blood [44].

Apnea hypopnea index and body mass index had a positive relationship (*r* = 0.24 and *p* < 0.05). Similar to our results, Pokharel M et al. reported in their study that body mass index and apnea hypopnea index were associated significantly (*p* = 0.010) [45]. The study found a direct association between obesity and OSA severity (*p* < 0.010), which is consistent with previously reported studies [46]. According to the Sleep Heart Health Study, people with the highest BMI had a prevalence rate of moderate-to-severe OSA that was three times higher than people with the lowest BMI [47]. Moreover, over a 4-year period, the Wisconsin Sleep Cohort Study revealed that for every 1% change in weight, OSA severity changed by about 3% [48]. According to a prior study, the severity of AHI was found to be correlated with all SpO2 measures (*p* < 0.001) [49]. Recent research suggests that the “hypoxic burden (HB)” specific to sleep apnea, which is the total of individual regions under the oxygen desaturation curve, may be useful in identifying high-risk OSA sufferers. In addition to the frequency of respiratory episodes, HB also captures OSA-related hypoxemia’s depth and duration, which may end up being crucial disease-characterizing characteristics. Last but not least, 15 min of 4% desaturations every hour seems to be able to spot patients who are more likely to have cardiovascular morbidity and mortality. So, treatment can alter the frequency of occurrences (for example, AHI) and possibly reduce the depth and length of desaturations, which may be sufficient to reduce the risk of undesirable health outcomes [50].

However, there is a biological reason for this variation in illness prevalence, including hormonal impacts and structural variations in the upper airways between the sexes [51].

Prior research has shown that abnormalities in the craniofacial anatomy have a role in the etiology of OSA and sleep apnea. Pharynx, glottis, and head-and-neck bone variations from the norm can make a person more susceptible to airway collapse and subsequent apneic episodes while they sleep [52].

## 5. Conclusions

In this study, a substantial prevalence of obesity and a high OSA frequency were found in the men. Our research showed that individuals with obstructive sleep apnea experience nocturnal desaturations. For better generalizability of the results, we recommend conducting future research with a larger sample size to include patients of all socioeconomic classes with or without comorbidities. Polysomnography is the primary test for early detection of this treatable condition. BMI, oxygen desaturation, snoring time, and number of snores showed significant correlations with AHI in the study group.

## 6. Future Recommendations

Future studies may establish that frequent arousals and nocturnal hypoxia are contributors to the emergence of cardiovascular illnesses. Therefore, we strongly advise adolescents to get screened for obesity and OSA. To prevent negative cardiovascular effects, awareness of OSA and obesity should be raised, especially in adolescents and young people.

## Figures and Tables

**Figure 1 jpm-13-00919-f001:**
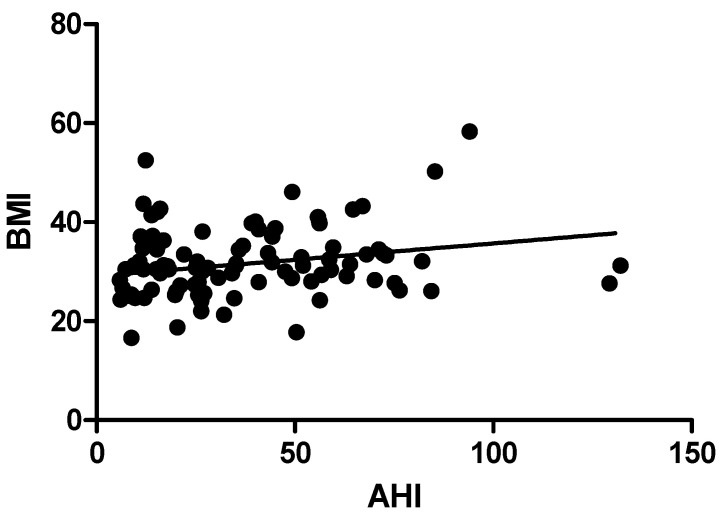
Correlation of Apnea Hypopnea Index with Body mass Index.

**Figure 2 jpm-13-00919-f002:**
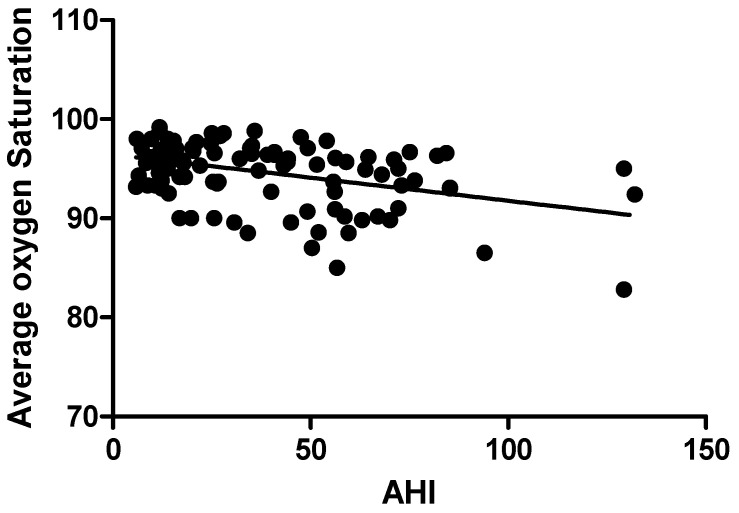
Correlation of Apnea Hypopnea Index with Average oxygen saturation.

**Figure 3 jpm-13-00919-f003:**
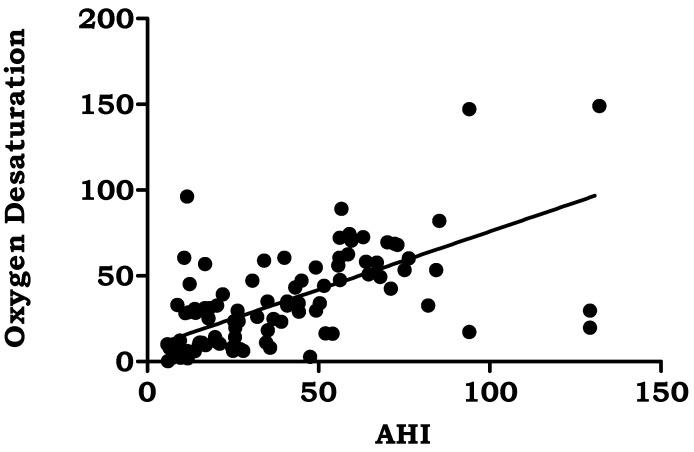
Correlation of Apnea Hypopnea Index with oxygen desaturation.

**Figure 4 jpm-13-00919-f004:**
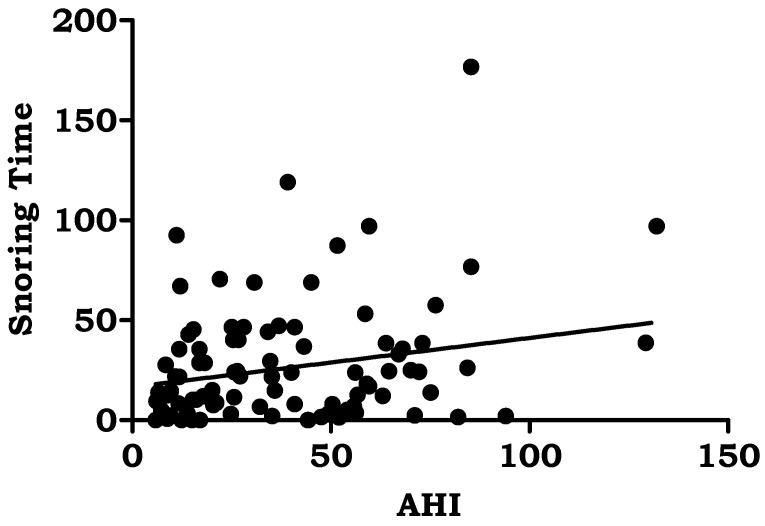
Correlation of Apnea Hypopnea Index with Snoring Time.

**Figure 5 jpm-13-00919-f005:**
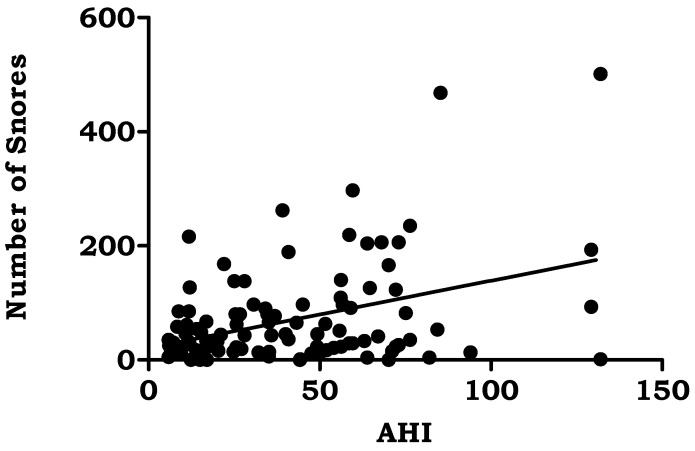
Correlation of Apnea Hypopnea Index with Number of snores.

**Table 1 jpm-13-00919-t001:** Demographic characteristics among study population.

Category	Non-OSA (41)	OSA Patients (175)
	Number (%)	Number (%)
Gender		
Male	32 (78.0)	143 (81.7)
Female	9 (22.0)	32 (18.3)
Age		
Male	55.41 ± 11.24	
Female	52.11 ± 3.12	

**Table 2 jpm-13-00919-t002:** Variables of study population.

Variables	Non-OSA (AHI < 5) *n* (%)	Mild OSA (AHI >5–15) *n* (%)	Moderate OSA (AHI16–29) *n* (%)	Severe OSA (AHI > 30) *n* (%)
AHI (events/hour)	41 (18.6)	30 (17.1)	60 (34.3)	85 (48.6)
BMI Kg/m^2^				
Ideal	15 (36.58)	6 (3.4)	7 (4)	6 (3.4)
Overweight	18 (43.9)	10 (5.7)	21(12)	21 (12)
Obese	8(19.5)	15 (8.6)	30 (17.1)	59 (33.7)
Smoking	5 (12.2)	4 (2.3)	14 (8)	17 (9.7)
Alcohol	6 (14.6)	7 (4)	15 (8.6)	33 (18.6)

**Table 3 jpm-13-00919-t003:** Analysis of variance among study population.

Variables	Non-OSA (AHI < 5)	Mild OSA (AHI 5–14.9)	Moderate OSA (AHI 15–30)	Severe OSA (AHI ≥30)	Significance (*p*)
AHI (events/hour)	1.69 ± 1.34	11.79 ± 3.55	22.12 ± 4.34	59.16 ± 22.15	0.00
BMI (kg/m^2^)	26.12 ± 2.72	31.66 ± 8.32	30.52 ± 3.99	34.35 ± 8.22	0.03
Average oxygen saturation (%)	95.92 ± 4.15	95.66 ± 1.98	95.22 ± 2.10	92.12 ± 3.44	0.00
Oxygen desaturation (events/hour)	6.09 ± 10.98	21.22 ± 20.66	23.38 ± 16.23	50.04 ± 26.67	0.00
Snoring time (minutes)	22.22 ± 39.61	21.99 ± 25.51	22.86 ± 16.49	31.39 ± 32.51	0.25
No. of snores	35.68 ± 59.78	41.49 ± 43.12	46.33 ± 39.58	86.44 ± 102.88	0.00

*p* < 0.05 significant, Post Hoc Tests—Scheffe.

**Table 4 jpm-13-00919-t004:** Symptoms among study population.

Symptoms	*n* (%)
Loud snoring	140 (80)
Breathing problem	98 (56)
Day time sleepiness	112 (64)
Family history of snoring	45 (25.7)

**Table 5 jpm-13-00919-t005:** Pearson correlation of AHI with variables in obstructive sleep apnea patients.

Variables	Mean ± SD	r-Value	*p*-Value
BMI (kg/m^2^)	32.17 ± 6.84	0.249	0.001
Total time analyzed (minutes)	417.03 ± 34.97	0.075	0.324
Average oxygen saturation (%)	94.65 ± 3.17	−0.387	0.000
Oxygen desaturation (events/hour)	29.29 ± 20.38	0.661	0.000
Snoring time (minutes)	24.48 ± 25.90	0.231	0.002
Number of snores	58.49 ± 62.86	0.383	0.000

*p* < 0.05 significant.

## Data Availability

The datasets used and analysed during the current study are available from the corresponding author on reasonable request.

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
