# Peer review of "A Prospective Study on Obstructive Sleep Apnea, Clinical Profile and Polysomnographic Variables"

_jpm, 2023, doi:10.3390/jpm13060919_

Round 1

Reviewer 1 Report

 The authors aim to assess the relationship between AHI and polysomnographic characteristics in OSA patients. A prospective study was conducted, 175 of them were reported to have OSA (AHI ≥ 5).  The authors found that BMI (r =0.249, p<0.001), average oxygen saturation (r=- 18 0.387, p<0.000), oxygen desaturation (r =0.661, p<0.000), snoring time (r =0.231, p<0.002), and number of snores (r = 0.383, p<0.001) were the polysomnographic variables that showed significant correlations with the AHI in the study group. There has been quite a number of research similar to this study in the English literature.  There are several major concerns that need to be addressed.

1. How were the confounding factors such as comorbid, smoking and other risk factors addressed? I would suggest a multi linear regression model

2. Why interpreted the PSG and the years of experience. How many technicians were involved.

3. Please highlight how this study adds to the current available knowledge

4. Please include future recommendations for this study

5. In conclusion, the authors mentioned: 'Our research showed that individuals with obstructive sleep apnea experience oxidative stress'. What do you mean by this?

 The authors aim to assess the relationship between AHI and polysomnographic characteristics in OSA patients. A prospective study was conducted, 175 of them were reported to have OSA (AHI ≥ 5).  The authors found that BMI (r =0.249, p<0.001), average oxygen saturation (r=- 18 0.387, p<0.000), oxygen desaturation (r =0.661, p<0.000), snoring time (r =0.231, p<0.002), and number of snores (r = 0.383, p<0.001) were the polysomnographic variables that showed significant correlations with the AHI in the study group. There has been quite a number of research similar to this study in the English literature.  There are several major concerns that need to be addressed.

1. How were the confounding factors such as comorbid, smoking and other risk factors addressed? I would suggest a multi linear regression model

2. Why interpreted the PSG and the years of experience. How many technicians were involved.

3. Please highlight how this study adds to the current available knowledge

4. Please include future recommendations for this study

5. In conclusion, the authors mentioned: 'Our research showed that individuals with obstructive sleep apnea experience oxidative stress'. What do you mean by this?

Author Response

[JPM] Manuscript ID: jpm-2364397

Dear Reviewers,

Thank you for your valuable comments. We have revised our article thoroughly as per your comments. The individual response to reviewer comments are given below.

Response to Reviewer(s)' Comments by Author(s):

Reviewer comment 1

The authors aim to assess the relationship between AHI and polysomnographic characteristics in OSA patients. A prospective study was conducted, 175 of them were reported to have OSA (AHI ≥ 5).  The authors found that BMI (r =0.249, p<0.001), average oxygen saturation (r=- 18 0.387, p<0.000), oxygen desaturation (r =0.661, p<0.000), snoring time (r =0.231, p<0.002), and number of snores (r = 0.383, p<0.001) were the polysomnographic variables that showed significant correlations with the AHI in the study group. There has been quite a number of research similar to this study in the English literature.  There are several major concerns that need to be addressed.

  1. How were the confounding factors such as comorbid, smoking and other risk factors addressed? I would suggest a multi linear regression model

Response 1

Thank you for this suggestion. We appreciate the reviewer’s feedback; However, the goal of the research was to assess OSA and the relationship between AHI and polysomnographic characteristics in OSA patients. We evaluated the Relationship between clinical and polysomnographic variables with Apnea Hypopnea Index (AHI) and Body Mass Index (BMI) were done using ANOVA and Pearson correlation coefficient]. Smoking, alcohol, and symptoms of the population were addressed through descriptive statistics and percentages.

Mentioned in Manuscript stated here:

20% of the patients in the current study were smokers, with 2.28% in the mild group, 8% in the intermediate group, and 9.71% in the severe group. Alcohol history revealed 31% of the population to be alcoholics, of whom 4% had mild OSA, 8.57% had moderate OSA, and 18.85% had severe OSA. Significant risk factors for OSA-related snoring include male gender, middle age, smoking, alcoholism, and physical infirmity (38). Based on previous studies, individuals with more severe OSA and more common CVD co-morbidities smoked more cigarettes per day, more packs per year, and consumed more Fagerstrom Test for Nicotine Dependence (39).

Each patient's symptoms were assessed, and 80% (n=140) of them reported loud snoring, 56% (n=98) had breathing issues, and 64% (n=112) reported feeling sleepy during the day. Among patients with obstructive sleep apnea, 25.71 percent of the population had a family history of snoring

Reference

Pokharel M, Shrestha BL, Dhakal A, Rajbhandari P, Shrestha KS, Kc AK, Bhattarai A, Karki DR. Clinical Profile and Diagnosis of Obstructive Sleep Apnea Syndrome using Overnight Polysomnography in a Tertiary Care Hospital. Kathmandu Univ Med J (KUMJ). 2021;19(75):361-365.

  1. Why interpreted the PSG and the years of experience. How many technicians were involved.

Response 2

Interpretation of  Polysomnogram (PSG) was done by the interventional pulmonologist and he has 10 years of experience. Two technicians were involved.

  1. Please highlight how this study adds to the current available knowledge

Response 3

Sleep disorders and sleep medicine are not recognized by both the general public and healthcare workers. According to this study, the men had a significant prevalence of obesity and a high frequency of OSA. The population of adults who are overweight or obese has to be reduced for the ideal body mass index for their healthy life. Excessive daytime sleepiness is the most common problem associated with OSA. Our study established that oxidative stress occurs in people with obstructive sleep apnea. The main test for early diagnosis of this curable condition is polysomnography which should be insisted to the public.

  1. Please include future recommendations for this study

Response 4  We have included the Future recommendations as stated by the reviewer

Future studies may establish that frequent arousals and nocturnal hypoxia are contributors to the emergence of cardiovascular illnesses. Therefore, we strongly advise adolescents to get screened for obesity and OSA. To prevent negative cardiovascular effects, awareness of OSA and obesity should be raised, especially among adolescents and young people through health education.

  1. In conclusion, the authors mentioned: 'Our research showed that individuals with obstructive sleep apnea experience oxidative stress'. What do you mean by this?

Response 5

Increased reactive oxygen species/reactive nitrogen species and oxidative stress are related with obstructive sleep apnea (OSA), which is characterized by intermittent hypoxia, and they have a negative impact on the associated cardio/cerebro-vascular disease.

Reference:

Lavie L. Oxidative stress in obstructive sleep apnea and intermittent hypoxia--revisited--the bad ugly and good: implications to the heart and brain. Sleep Med Rev. 2015 Apr;20:27-45. doi: 10.1016/j.smrv.2014.07.003. Epub 2014 Jul 24. PMID: 25155182.

Once again, thank you to all the reviewers for the valuable comments and suggestions. We have revised our manuscript strictly and thoroughly as per your comments for more clarity to the reader.

Thank you

Geetha Kandasamy

Reviewer 2 Report

This study aimed to investigate the relationship between obstructive sleep apnea (OSA) and polysomnographic characteristics in OSA patients. The study was conducted over a two-year period and included 216 participants who underwent polysomnography, with 175 of them reported to have OSA. The results showed that OSA affects 6.3% of males and 1.4% of females, and the average age of the study group was 53.77 ± 7.19. The study found that individuals with severe OSA had 59.16 ± 22.15 events/hour. The study also found that BMI, average oxygen saturation, oxygen desaturation, snoring time, and the number of snores were polysomnographic variables that showed significant correlations with the AHI in the study group. The study concludes that a substantial prevalence of obesity and high OSA frequency was found in 6.3% of the men, and early detection of OSA through polysomnography is essential.

Overall, this study provides important insights into the prevalence of OSA and its relationship with polysomnographic characteristics, emphasizing the importance of early detection of this condition through polysomnography. The study's findings also suggest that obesity is a significant risk factor for OSA, which highlights the importance of maintaining a healthy weight to prevent OSA. The study is well-designed, and the results are presented in a clear and concise manner. However, the study has limitations, such as the relatively small sample size, which may affect the generalizability of the results. Nonetheless, this study provides valuable information for healthcare professionals who are involved in the management of OSA patients.

Author Response

[JPM] Manuscript ID: jpm-2364397

Dear Reviewers,

Thank you for your valuable comments. We have revised our article thoroughly as per your comments. The individual response to reviewer comments are given below.

Response to Reviewer(s)' Comments by Author(s):

Reviewer comment 2

This study aimed to investigate the relationship between obstructive sleep apnea (OSA) and polysomnographic characteristics in OSA patients. The study was conducted over a two-year period and included 216 participants who underwent polysomnography, with 175 of them reported to have OSA. The results showed that OSA affects 6.3% of males and 1.4% of females, and the average age of the study group was 53.77 ± 7.19. The study found that individuals with severe OSA had 59.16 ± 22.15 events/hour. The study also found that BMI, average oxygen saturation, oxygen desaturation, snoring time, and the number of snores were polysomnographic variables that showed significant correlations with the AHI in the study group. The study concludes that a substantial prevalence of obesity and high OSA frequency was found in 6.3% of the men, and early detection of OSA through polysomnography is essential.

Overall, this study provides important insights into the prevalence of OSA and its relationship with polysomnographic characteristics, emphasizing the importance of early detection of this condition through polysomnography. The study's findings also suggest that obesity is a significant risk factor for OSA, which highlights the importance of maintaining a healthy weight to prevent OSA. The study is well-designed, and the results are presented in a clear and concise manner. However, the study has limitations, such as the relatively small sample size, which may affect the generalizability of the results. Nonetheless, this study provides valuable information for healthcare professionals who are involved in the management of OSA patients.

Response: We hope that Reviewer 2 is satisfied with the manuscript and thank you for his/her valuable comments.

We have addressed all typos and English grammar in the entire manuscript

Once again, thank you to all the reviewers for the valuable comments and suggestions. We have revised our manuscript strictly and thoroughly as per your comments for more clarity to the reader.

Thank you

Geetha Kandasamy

Round 2

Reviewer 1 Report

The authors have addressed the revision adequately

None

Author Response

Dear Reviewer

Thank you for your valuable comments. We have revised our article thoroughly as per your comments.